# Research on Landing Stability of Four-Legged Adaptive Landing Gear for Multirotor UAVs

**Xinlei Ni** [1,2], **Qiaozhi Yin** [1,2,3], **Xiaohui Wei** [1,2,3,*], **Peilin Zhong** [1,2] **and Hong Nie** [1,2,3]

1   State Key Laboratory of Mechanics and Control of Mechanical Structures, Nanjing University of Aeronautics and Astronautics, Nanjing 210016, China
2   Key Laboratory of Fundamental Science for National Defense-Advanced Design Technology of Flight Vehicle, Nanjing University of Aeronautics and Astronautics, Nanjing 210016, China
3   National Key Laboratory of Rotorcraft Aeromechanics, Nanjing University of Aeronautics and Astronautics, Nanjing 210016, China
*   Correspondence: wei_xiaohui@nuaa.edu.cn

**Abstract:** Rotorcraft Unmanned Aerial Vehicles (UAVs) often need to take off and land under complex working conditions. The rugged terrains may cause the UAV to tilt during takeoff and landing and even cause rollover and other accidents in severe cases. In this paper, a new four-legged landing gear of multirotor UAVs with a passive cushioning structure is designed, aiming at the landing stability requirement of rotorcraft UAVs in complex terrains. The mathematical model of the landing gear dynamics is established in MATLAB/Simulink, and the drop test simulation is carried out under different landing terrain conditions. By comparing the simulation results of the drop test multibody dynamic model in Simcenter3D dynamics software, the adaptive landing and cushioning capacity of the landing gear and the accuracy of the mathematical model are verified. Combined with the landing stability criterion and control strategy of adaptive landing gear adjustment, the landing stability of adaptive landing gear under different slope angles of landing surface and horizontal velocities is studied. The landing stability boundary under different combinations of these two parameters is found.

**Keywords:** adaptive landing gear; multibody dynamics; drop test simulation; landing stability

## 1. Introduction

Due to their vertical takeoff and landing characteristics, rotorcraft Unmanned Aerial Vehicles (UAVs) have played an important role in field exploration, border patrol, and other scenarios. Traditional landing gear of UAVs is mainly truss type which is relatively fixed. The landing stability is difficult to ensure when taking off and landing on uneven ground and may cause the fuselage to roll over [1]. In the face of unexpected tasks in complex natural environments, it is often necessary to return to a fixed location to complete the takeoff and landing process, which seriously limits its scope of use [2]. Therefore, it is necessary to carry out research on the new landing structure design to meet the landing stability requirement of rotorcraft UAVs in complex terrains.

Based on the concept of adaptive landing and bionics, a variety of adaptive landing gear structures suitable for rotorcraft UAVs were proposed, which could adapt to different terrains by adjusting the landing gear structure actively and enhancing the landing stability. Boix [3] proposed an articulated landing gear, which can adapt to different terrains by driving motors on outriggers; Stolz [4] proposed a four-legged single-joint linkage landing gear with a simple structure and strong bearing capacity. Liu [5] proposed a four-finger multijoint adaptive landing system, which made the landing gear fingers curl up through a single motor and adapt to the ground. It had a multistage transmission structure and strong adaptability to terrains. Roderick [6] and Hang [7] proposed landing gear imitating a bird's claw, which allowed the aircraft to stay stable in special places such as tree branches and

grab objects. To better meet the landing stability requirement of rotorcraft UAVs in complex terrains, it is necessary to propose an adaptive landing gear structure with a lightweight, simple structure, large bearing capacity, and strong terrain versatility.

The landing process of adaptive landing gear in difficult terrains is similar to the landing of vertical landing vehicles such as lunar landers. Many studies have been carried out on the landing dynamics and landing stability of such vertical landing vehicles. In the 1960s, the United States proposed that the four legs were the best landing structure in terms of stability and weight by building a multibody dynamic model for dynamic simulation and experimental methods [8]. Witte [9,10] conducted a landing dynamics simulation of Rosetta lander Philae and established a landing safety look-up table and landing safety assessment method. Yang [11] used the energy method to determine whether the landing device had overturned so as to carry out the overall scheme design of the landing leg. Nie [12,13] used transient dynamics to simulate the landing impact process considering the flexibility of the lander and the ground. Deng [14,15] gave a modeling method of landing dynamics for a four-legged truss lunar lander and introduced the zero-moment-point method into the landing stability analysis. Compared with the above aircrafts, the adaptive landing gear can be adjusted according to the terrain conditions, so the influence of factors such as landing forms and attitude angles on its landing stability can no longer be considered. The relationship between other external parameters and landing stability should be studied.

For multirotor UAVs that need to take off and land on complex terrains, a new four-legged adaptive landing gear with a passive cushioning structure is designed, and its landing stability is studied. In Section 2, the bionic configuration and operating principle of the four-legged landing gear are elaborated. Further, dynamic models of the landing gear are established in MATLAB/Simulink and Simcenter3D, respectively. In Section 3, the drop test simulations under different landing terrains are carried out for the two models, and the adaptive landing and cushioning capacity of the landing gear and the accuracy of the mathematical model are verified by comparing the drop test results of the models in MATLAB/Simulink and Simcenter3D. In Section 4, the landing stability of adaptive landing gear with different slope angles and horizontal velocities is studied, combined with the landing stability criterion and control strategy of adaptive landing gear adjustment, and the landing stability boundary with combinations of two parameters is found. Conclusions are drawn in Section 5.

## 2. Dynamic Modeling of Adaptive Landing Gear

### 2.1. Structure Design

When designing the landing gear of traditional rotorcraft, the main consideration is the cushioning performance of the landing gear, landing stability, and the proportion of the landing gear's weight. In order to achieve active adaptability to different terrains, adaptive landing gear needs to increase the drivers to meet the demand, resulting in an increase in structural complexity and weight. To meet the requirements of adaptive landing gear, this paper chooses the bionic four-legged structure, which provides a certain stability margin and improves the bearing capacity while landing. Each leg has three motors and can move the foot in the three-dimensional working space, which meets the adaptability to different terrains.

In order to reduce the load of adaptive landing gear during takeoff and landing, a passive cushioning structure is designed to imitate the mechanism of the human foot and ankle. As shown in Figure 1a, the human foot and ankle are composed of bones, muscles, tendons, ligaments, etc., and are the only joints that interact with the ground during most human movements [16]. In modern research on ankle biomechanics and ankle pedaling mechanism, muscles and tendons are regarded as elastic elements such as springs, which store and dissipate energy through elastic bending and stretching [17]. Therefore, a separate passive joint (Ankle joint) is designed in addition to the three active joints of the leg to imitate the human ankle, and a slant shock absorber is installed to imitate the muscle tendons between the instep and the calf. The passive joint and the shock absorber form

a bionic cushioning structure of the landing gear leg, which can absorb and dissipate the impact energy of the falling and reduce the load on the landing gear.

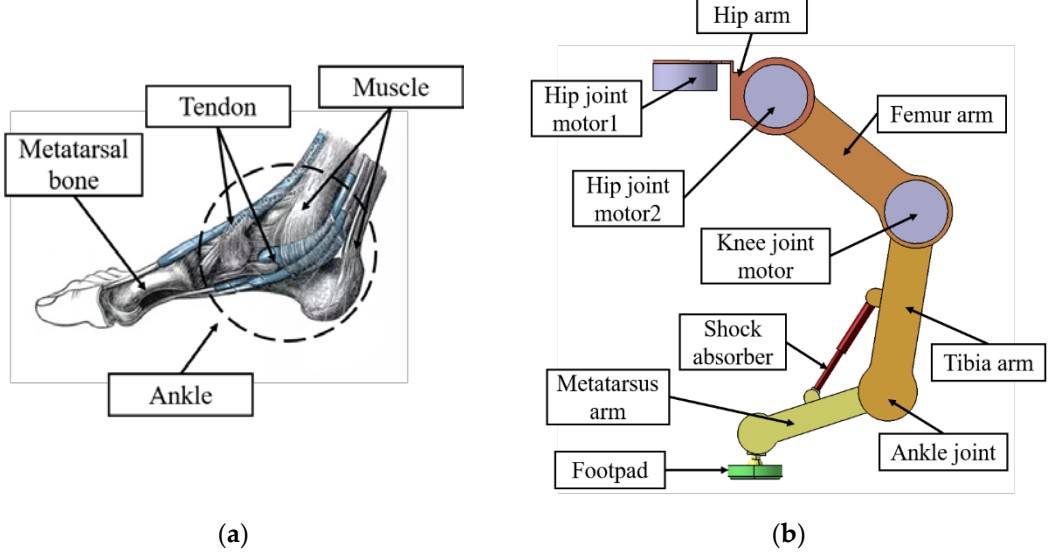

|  (**a**)  |  (**b**)  |

**Figure 1.** Comparative diagram of the human foot and landing leg: (**a**) represents the anatomical view of the foot and ankle; (**b**) represents the structure of a single leg of the adaptive landing gear.

The landing gear structure is modeled in Computer Aided Three-dimensional Interactive Application (CATIA), and its single-leg model is shown in Figure 1b. The three brushless motor joints and one passive joint are named: Hip joint 1, Hip joint 2, Knee joint, and Ankle joint. The connecting arms between each joint are named: Hip arm, Femur arm, Tibia arm, and Metatarsus arm. A Footpad is installed at the end of the Metatarsus arm through a ball hinge to increase the contact area with the ground.

In order to highlight the focus of this paper, the following simplifications are made for the drop test simulation model of a whole multirotor UAV: Ignore the influence of rotor aerodynamics on the movement of the airframe. Omit the interstructural connectors, and the connections between the components are reliable. Place nonmajor components such as controllers and power supplies as a mass block inside the fuselage. A virtual prototype model of the multirotor UAV is built in CATIA. The mass of the multirotor UAV and each component is shown in Table 1. The number of each landing leg (1, 2, 3, and 4) and the body axes $(O_b, X_b, Y_b,$ and $Z_b)$ are shown in Figure 2.

**Table 1.** The table shows the mass and number of each component and the mass of the multirotor UAV.

| Component | Number | Mass (Each)/kg |
| --- | --- | --- |
| Fuselage | 1 | 16 |
| Hip arm | 4 | 0.2 |
| Femur arm | 4 | 0.4 |
| Tibia arm | 4 | 0.4 |
| Metatarsus arm | 4 | 0.3 |
| Shock absorber | 4 | 0.1 |
| Footpad | 4 | 0.1 |
| Motor | 12 | 0.9 |
| Total |  | 32.8 |

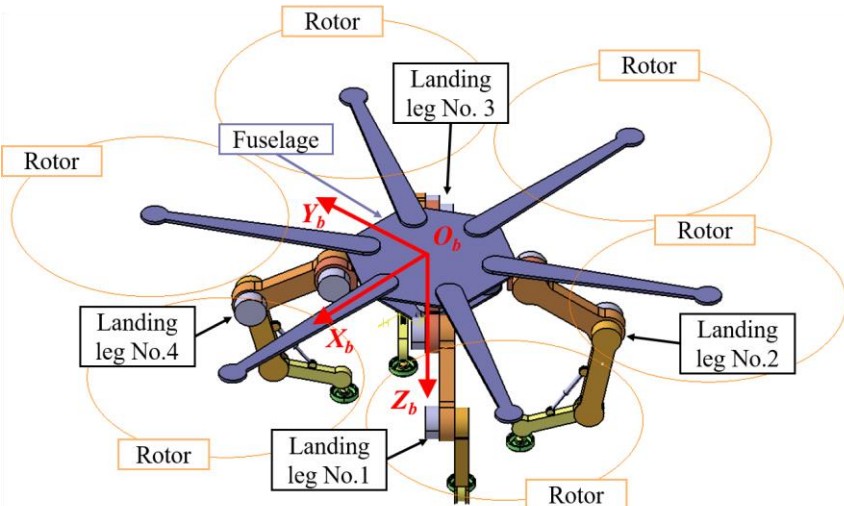

**Figure 2.** The virtual prototype model of the multirotor UAV in CATIA.

*2.2. Dynamic Modeling*

Modeling landing gear dynamics in MATLAB/Simulink requires dynamic analysis of the entire landing gear machine. During the drop test simulation, the multirotor UAV ignores the influence of aerodynamics. The torque and force on the landing gear during landing can be expressed as:

$$\begin{bmatrix} \tau_b \\ F_b \end{bmatrix} = \begin{bmatrix} \sum_{i=1}^{4} r_i \times F_i \\ G + \sum_{i=1}^{4} F_i \end{bmatrix}, \tag{1}$$

where $r_i$ is a vector from the footpad to the center of mass of the multirotor UAV, $G = \begin{bmatrix} 0 & 0 & mg \end{bmatrix}^T$ is the gravity vector, and $F_i = \begin{bmatrix} F_{\mu x} & F_{\mu y} & F_n \end{bmatrix}$ is a vector of tangential friction and normal impact of each landing legs' footpad.

In order to describe the normal collision process and energy dissipation between the landing gear and the ground, the Kelvin–Voigt equivalent continuous contact force model is introduced to establish the normal impact at the footpad of the landing gear [18]. The direction of impact force is perpendicular to the ground upward, and the basic form of the model is:

$$F_n = K \cdot \delta^{\frac{3}{2}} + D \cdot \dot{\delta}, \tag{2}$$

where $K$ is the Hertz contact stiffness coefficient, $D$ is the damping coefficient, $\delta$ is the relative deformation, and $\dot{\delta}$ is the relative collision velocity.

The force of friction between the landing gear and the ground adopts the Stribeck model [19,20], which determines the nonlinear friction based on the relative speed of motion. The direction of friction is parallel to the ground and opposite to the relative velocity between the footpad and the ground. The friction and coefficient of friction can be expressed as:

$$\begin{cases} F_\mu = F_n \cdot \mu(v) \\ \mu(v) = \mu_s \cdot sgn(v) - k_1 \cdot v + k_2 \cdot v^3, \end{cases} \tag{3}$$

where $F_n$ is the normal force, $v$ is the velocity of relative motion, $k_1$ and $k_2$ are constants related to the static friction coefficient $\mu_s$, the kinetic friction coefficient $\mu_m$, and the minimum kinetic friction speed $v_m$.

Assuming that the shock absorber model used is a spring-damping model, the cushioning force can be expressed as:

$$F_m = K \cdot \delta + D \cdot \dot{\delta}, \tag{4}$$

where $K$ is the stiffness coefficient of the shock absorber, $D$ is the damping coefficient, $\delta$ is the cushion stroke, and $\dot{\delta}$ is the relative speed of the shock absorber.

All the force vectors in Equations (1)–(4) are shown in Figure 3.

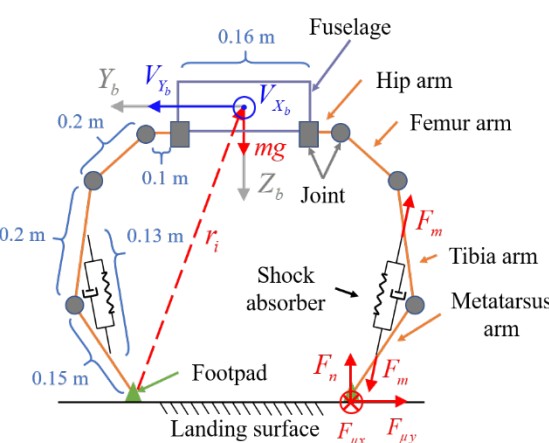

**Figure 3.** Diagram of aircraft component dimensions and vectors in Equations (1)–(4). The force vectors are shown in red, velocity vectors are shown in blue, and body axes are shown in gray.

## 3. Drop Test Simulation of Adaptive Landing Gear

Based on the mathematical model of adaptive landing gear dynamics in Section 2, the adaptive landing gear landing dynamic models are established in MATLAB/Simulink and Simcenter3D dynamics software, respectively. The drop test dynamic simulation under complex terrains of adaptive landing gear is carried out, and the framework of motion pairs is shown in Figure 4.

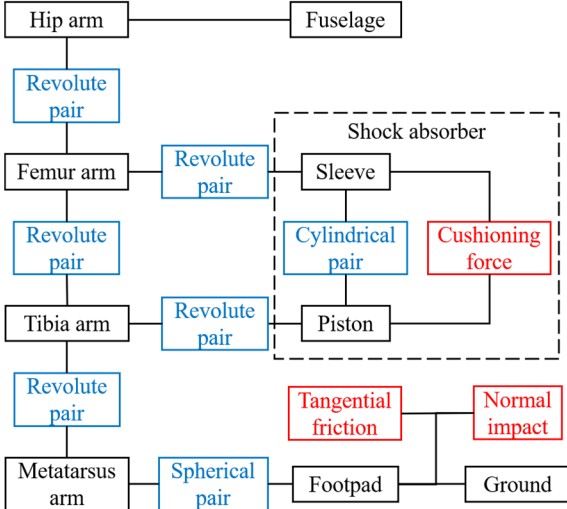

**Figure 4.** The framework of motion pairs in the dynamic model of the adaptive landing gear.

### 3.1. Control Strategy of Adaptive Landing Gear Adjustment

The topographic data of complex terrains (such as slopes, pits, etc.) can be described by the superimposed use of relative height difference and relative slope angle. In the drop test simulation, the landing terrain is simplified as a surface only with height difference or slope angle. The drop test ground models with a height difference of 0.05 m and slope angle of 10° are established, as shown in Figure 5.

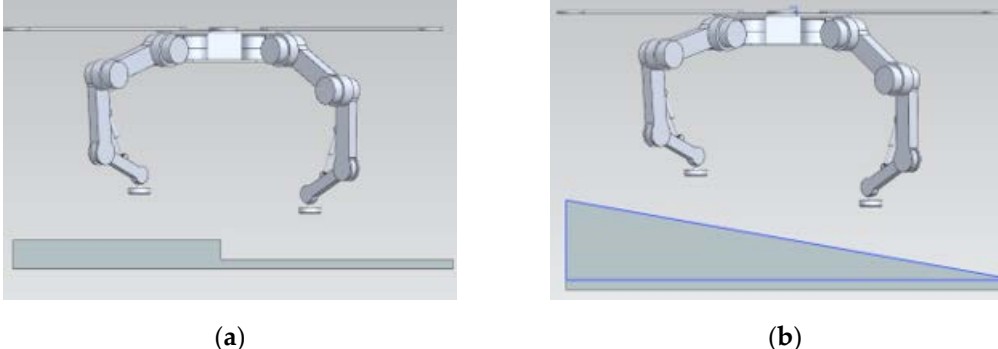

**Figure 5.** The simplified landing terrains with height difference or slope angle in Simcenter3D: (**a**) represents a drop test ground model with a height difference of 0.05 m; (**b**) represents a drop test ground model with a slope angle of 10°.

Based on the simplified drop test ground models above, a control strategy of landing legs' adjustment is proposed. The target of landing legs' adjustment is to make all four legs' footpads touch the ground at the same time to bear the load, avoiding '1-2-1', '2-2', and other landing forms which may bear large loads. The main plane of the rotorcraft remains parallel to the horizontal plane of the world frame during hovering, and the coordinates of the four landing legs' footpads in the body axes are $(\pm W, \pm L, H)$. The relative height difference along the $Z_b$-axis obtained by the sensors on the landing legs is $2\Delta H$. Keep the relative positions of the four legs' footpads along the $X_b$-axis and $Y_b$-axis the same so that the height difference obtained does not change. Change the relative positions of the landing legs along the $Z_b$-axis to $(\pm W, \pm L, H \pm \Delta H)$, adapting to the landing terrains with height difference or slope angle. In this paper, the data are set as $W = 0.2$ m, $L = 0.3$ m, and $H = 0.4$ m. Because of the limitation of the adaptive landing gear structure, the maximum adaptable height difference is $\Delta H = \pm 0.1$ m, and the maximum adaptable slope angle is $\alpha = \pm 20°$. The schematic diagram of the control strategy is shown in Figure 6.

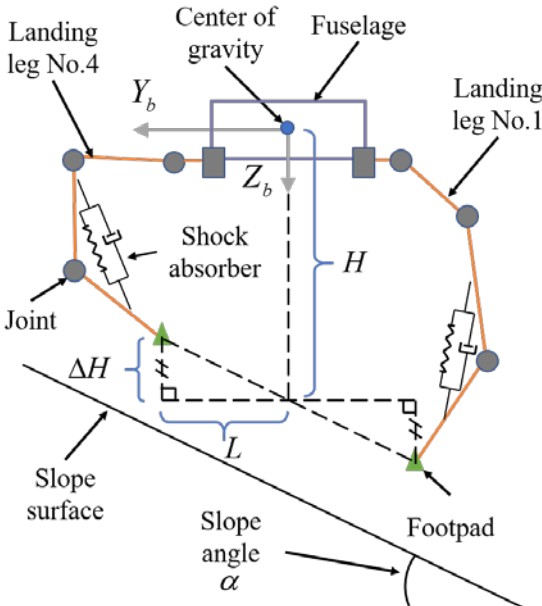

**Figure 6.** Schematic diagram of the control strategy for adjusting landing legs to adapt to complex landing terrains.

*3.2. Simulation Analysis of Seismic Drops*

In MATLAB/Simulink and Simcenter3D, the rotorcraft has been set as the state adapting to the landing terrain. The rotorcraft starts free fall and land on the surfaces with

a height difference of 0.05 m and slope angle of 10°, respectively, at a vertical speed of 1.5 m/s. The ground model used is symmetrical along the $Y_b O_b Z_b$ plane of the body axes, so the drop test process mainly causes the change in the roll angle among all attitude angles. Define that the initial vertical velocity is positive along the positive direction of $Z_b$-axis. The roll angle is positive along the $X_b$-axis. The normal force on the footpad is positive when perpendicular to the landing surface upward. The cushion stroke is positive for compression. The starting height of the free fall can be calculated by $h = v^2/(2 \cdot g)$.

The drop test simulation curves in different software in different terrains are shown in Figure 7. The rotorcraft is in free fall for about 0.15 s from the starting height and touches the ground at a speed of 1.5 m/s. The vertical velocity decreases rapidly, reaching the negative extreme value at around 0.17 s and converging to 0 after several fluctuations, as shown in Figure 7a. The roll angle of the fuselage has a large convex peak at 0.15–0.2 s and then gradually increases to a small steady-state value, as shown in Figure 7b. The normal force of the footpad of the landing leg No. 1 reaches a load peak of about 2400–2600 N at about 0.15 s and quickly decreases to 0. After the second load peak, the normal force converges to a steady-state value, as shown in Figure 7c. The cushion stroke of the shock absorber on landing leg No. 1 increases abruptly at 0.15 s and then decreases briefly. At about 0.28 s, the cushion stroke begins to gradually increase to the steady-state value, as shown in Figure 7d.

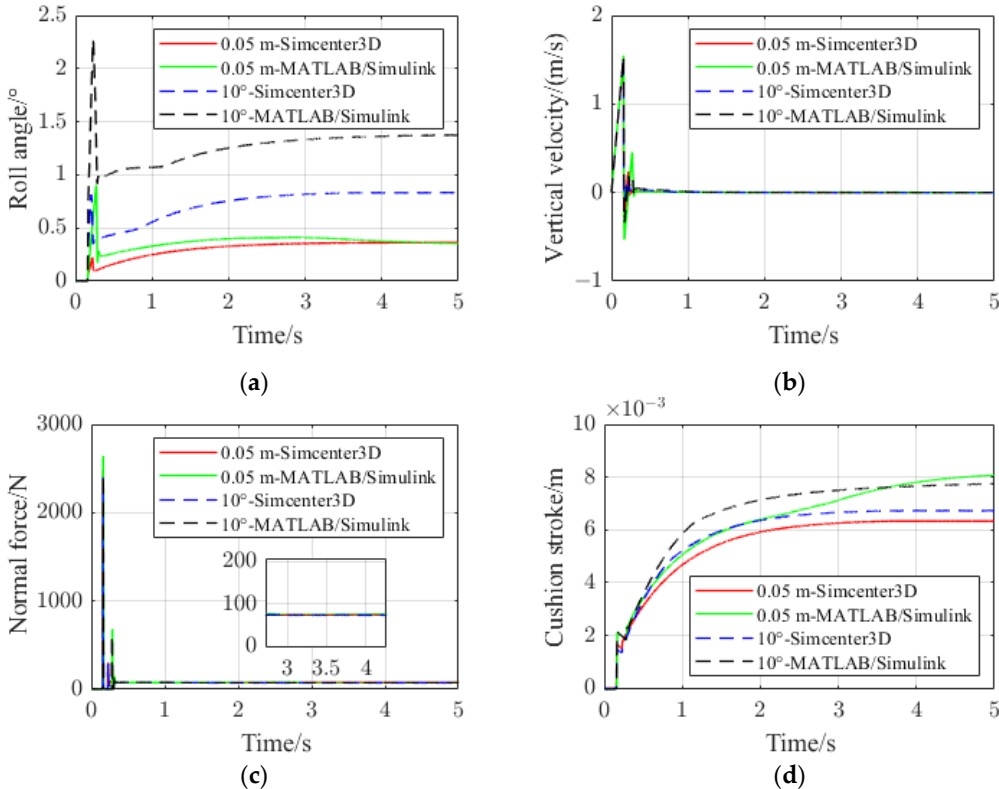

**Figure 7.** The drop test simulation curves in MATLAB/Simulink and Simcenter3D in landing terrains with a height difference of 0.05 m and slope angle of 10°: (**a**) represents the roll angle of the whole rotorcraft; (**b**) represents the vertical velocity of rotorcraft; (**c**) represents the normal force acting on the footpad of landing leg No. 1; (**d**) represents the cushion stroke of the shock absorber on landing leg No. 1.

The analysis of Figure 7 shows the motion of the adaptive landing gear in the drop-test simulation. After the landing gear touches the ground at a vertical speed of 1.5 m/s, the shock absorbers quickly compress and dissipate energy. Because of the excessive vertical speed and the large normal force at the footpad, the landing gear rebounds after touching the ground, resulting in a negative vertical velocity. The footpad leaves the ground, and the

cushion stroke of the shock absorber restores. Because of the adjustment to the adaptive landing gear, the moments of inertia of the fuselage change slightly, and the fuselage rolls in the air. After the landing gear falls to the ground again, the normal force of the footpad reaches a second peak, and the roll angle of the fuselage increases to the extreme value and decreases rapidly. Then, the vertical velocity and normal force quickly converge to the steady-state value while the roll angle and cushion stroke slowly increase to the steady-state value. In the state adapting to the landing terrain, the angles between each shock absorber and the ground are different, and the steady-state cushion strokes are different, which makes ankle joints of each leg rotate relative to the fuselage and produce a steady-state value of roll angle.

Compared with the drop test simulation curves in landing terrains with a height difference of 0.05 m and slope angle of 10°, it can be seen that the normal force peak value and roll angle are larger when the landing gear lands on the terrain with a height difference of 0.05 m. According to the control strategy shown in Figure 6, the angles between each shock absorber and the ground are different when the landing gear lands on different terrains, so the steady-state value of the cushion strokes and the changes in roll angle are different. When the landing gear lands on a slope of 10°, its normal velocity component perpendicular to the slope is less than 1.5 m/s, and the normal force at the footpad is positively correlated with the relative collision velocity in Formula (2). Therefore, the resultant footpad's normal force is small.

Through the drop test simulations in different terrains above, it is verified that the structure of adaptive landing gear stabilizes the speed and attitude of the rotorcraft in a short time during landing and has the adaptive landing capacity in complex terrain conditions.

Comparing the drop test simulation curves in MATLAB/Simulink and Simcenter3D, it can be seen that when the landing gear lands on terrain with a height difference of 0.05 m, the peak normal force is 2644 N in MATLAB/Simulink and 2471 N in Simcenter3D, with an error of 7%. The difference between the steady-state value of the cushion stroke is 0.002 m, and the difference between the steady-state value of the roll angle is 0.01°. When the landing gear lands on terrain with a slope angle of 10°, the peak normal force is 2427 N in MATLAB/Simulink and 2399 N in Simcenter3D, with an error of 1%. The difference between the steady-state value of the cushion stroke is 0.001 m, and the difference between the steady-state value of the roll angle is 0.5°.

From the movement of the adaptive landing gear above, it can be seen that the changes in the vertical velocity, cushion stroke, and roll angle are all related to the normal force acting on the footpad of the landing leg. The maximum error of the peak value of the normal force is 7% under the two terrain conditions, and the error is acceptable considering the simplification of the structural parameters in the mathematical model. The mathematical model of dynamics is accurate enough to be used to conduct subsequent research on the landing stability of adaptive landing gear.

## 4. Research on Landing Stability

Based on the mathematical model of landing dynamics and the control strategy in complex terrains in Section 3, different initial parameters related to the landing process are selected to search for the landing stability boundary of the adaptive landing gear. When the aircraft lands on the slope surface, the normal impact force generated between the footpad and ground can be deconstructed into a vertical component and a horizontal component, and the horizontal component of impact force may make the rotorcraft tend to be unstable. Similarly, the horizontal speed of the rotorcraft caused by sudden crosswinds during landing will also make the rotorcraft tend to be unstable. The different landing attitude angles of the fixed configuration vertical takeoff and landing carriers may produce different landing forms such as '1-2-1' and '2-2'. While the adaptive landing gear can adjust the four legs to touch the ground at the same time, the landing attitude angles do not affect the landing form. Therefore, in the search for the landing stability boundary of adaptive landing gear in this paper, the influences of the initial horizontal velocity and slope angle

of the landing terrain are mainly considered. Define the initial horizontal velocity $V_Y$ as positive along the positive direction of $Y_b$-axis. The slope angle $\alpha$ is positive, as shown in Figure 6.

### 4.1. Landing Stability Criterion

During the search for the landing stability boundary, the criterion of stability distance is used to determine whether the rotorcraft has overturned [21]. As shown in Figure 8, points $P_1 - P_4$ show the coordinates of the four legs' footpads in the world frame. The point $P_m$ shows the coordinate of the center of gravity of the rotorcraft. Points $P_1' - P_4'$ and $P_m'$ are the projections of the corresponding points on the $XOY$ plane of the world frame. The distances from $P_m'$ to each side of the quadrangle $P_1'P_2'P_3'P_4'$ are shown:

$$d_1 = \frac{\overrightarrow{P_1'P_2'} \times \overrightarrow{P_1'P_m'}}{\left|\overrightarrow{P_1'P_2'}\right|}, d_2 = \frac{\overrightarrow{P_2'P_3'} \times \overrightarrow{P_2'P_m'}}{\left|\overrightarrow{P_2'P_3'}\right|}, d_3 = \frac{\overrightarrow{P_3'P_4'} \times \overrightarrow{P_3'P_m'}}{\left|\overrightarrow{P_3'P_4'}\right|}, d_4 = \frac{\overrightarrow{P_4'P_1'} \times \overrightarrow{P_4'P_m'}}{\left|\overrightarrow{P_4'P_1'}\right|} \tag{5}$$

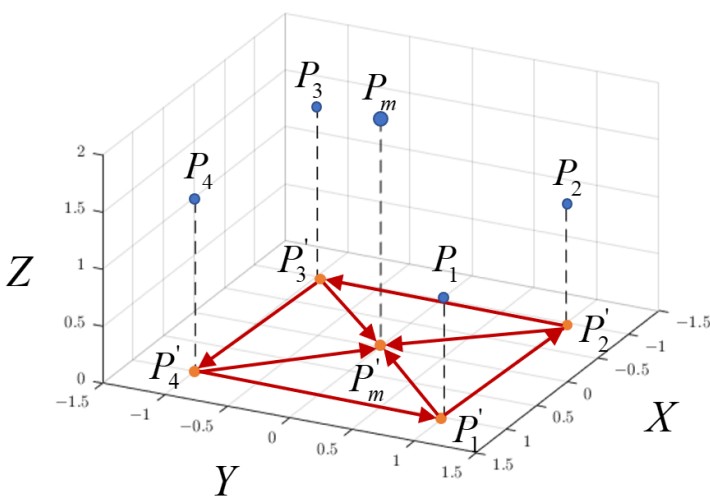

**Figure 8.** The projections of feature points in the world frame.

The stability distance is defined as the minimum value of the distances above. When the stability distance is greater than 0, that is $d_{min} = min(d_1, d_2, d_3, d_4,) > 0$, the projection of the center of gravity is inside the range of the quadrangle $P_1'P_2'P_3'P_4'$, which means the rotorcraft has not overturned.

### 4.2. Landing Stability Simulation

In order to verify the landing process under different initial parameters, two working conditions are selected as examples for simulation verification. At the same time, a mathematical model of the conventional four-corner-fixed landing gear is established as a contrast. The positions of four fulcrums in body axes are $(\pm W, \pm L, H \pm \Delta H)$ and nonadjustable. Set the vertical velocity is 1.5 m/s along the positive direction of $Z_b$-axis and other initial parameters are set as follows:

1. Slope angle $\alpha = 0°$, horizontal speed $V_Y = 1.5$ m/s;
2. Slope angle $\alpha = 10°$, horizontal speed $V_Y = -0.2$ m/s.

The landing simulation curves of adaptive landing gear and conventional landing gear are shown in Figure 9. The roll angles of the adaptive landing gear have certain fluctuations after landing under two parameter settings and stabilize after 0.7 s and 0.2 s, as shown in Figure 9a. The adaptive landing gear maintains stable distances greater than 0 at both parameter settings, as shown in Figure 9b. The stability distances indicate that

the landing gear does not overturn and show the same trend as the curve of roll angle, indicating the feasibility of the landing stability criterion. The stability distances of the conventional landing gear under two parameter settings change to less than 0 at 1.8 s and 0.4 s, indicating that the landing gear has overturned. According to the parameters of the conventional landing gear, it is clear that the gravity of the landing gear creates a torque that aggravates the overturning of the rotorcraft when roll angle $|\varphi| \geq \tan^{-1}(L/H)$. As shown in Figure 9a, the roll angles grow to greater than $\tan^{-1}(L/H)$ at 1.8s and 0.4s and keep growing to 180°. The roll angles indicate that the conventional landing gear has overturned and show the same trend as the stability distance curves, also indicating the feasibility of the landing stability criterion. Based on the data in Figure 9, the adaptive landing gear has better landing stability than the conventional landing gear under the same landing conditions and initial parameters.

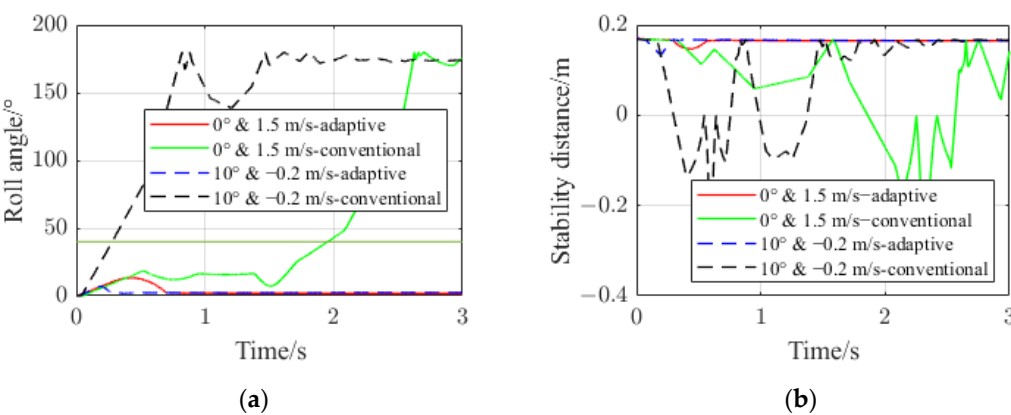

(a)                                           (b)

**Figure 9.** The landing simulation curves of adaptive landing gear and conventional landing gear with different combinations of vertical velocity and slope angle: (**a**) represents the roll angle of the rotorcraft; (**b**) represents the stability distance.

### 4.3. Landing Stability Boundary

On the basis of the mathematical model in MATLAB/Simulink, stability criterion, and the control strategy in complex terrains, the landing stability boundary in different initial parameter combinations is searched by cyclically changing the slope angle and initial horizontal velocity. Because of the limitation of the adaptive landing gear structure, the maximum adaptable slope angle is ±20°. Considering the symmetry of the positive and negative slope angles, set the range of slope angle $\alpha$ to 0–20°. Set the vertical velocity is 1.5 m/s along the positive direction of $Z_b$-axis. Set the range of initial horizontal speed $V_Y$ to −15–15 m/s, which is in the cruising speed range of most multirotor UAVs (<20 m/s). Due to space constraints, the figure only shows the horizontal speed range of −10.5–2 m/s, and the variation tendency of the range not shown is the same. The landing stability boundary in different combinations of slope angle and initial horizontal velocity is shown in Figure 10.

The stability boundary of adaptive landing gear is shown in Figure 10. The white area in the figure is the stable landing area, indicating that the landing gear can land stably at a vertical velocity of 1.5 m/s at the corresponding horizontal velocity and slope angle of the terrain. Because of the limitation of the adaptive landing gear structure, the maximum adaptable slope angle is 20° as the limit of the yellow area shown in the figure. When the landing gear lands on the landing surface at a vertical velocity of 1.5 m/s, the rotorcraft may overturn with certain combinations of the horizontal velocity and slope angle, as the blue area shown in the figure.

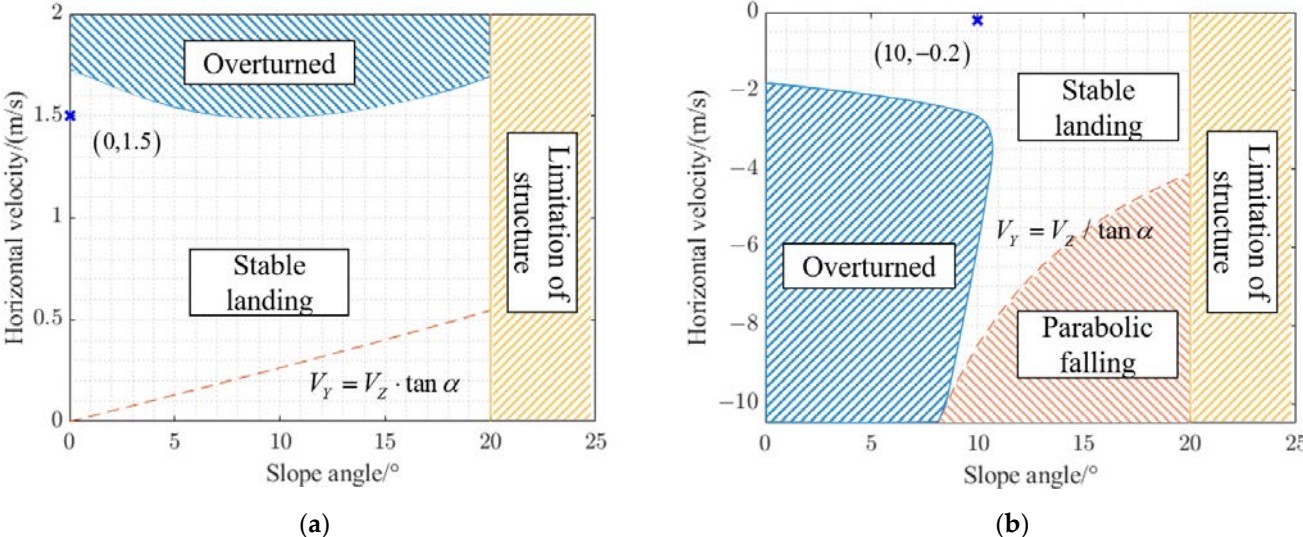

**Figure 10.** The landing stability boundary considering the influences of initial vertical velocity and slope angle: (**a**) represents boundary with positive velocity and slope angle varying in the range of 0–20°; (**b**) represents boundary with negative velocity and slope angle varying in the range of 0–20°.

The red dashed line in Figure 10a is used to determine the direction o of the relative motion trend after the rotorcraft contacts with the slope, and the points on it are still in the stable landing area. When the combinations of horizontal velocity and slope angle are above the red dashed line, the direction of relative motion trend is along the slope upward. The red area in Figure 10b indicates that the horizontal velocity is too large, and the rotorcraft is not in direct contact with the landing surface at the beginning of the simulation. When the horizontal velocity is too large, the rotorcraft falls in a parabolic trajectory, and its vertical velocity is greater than 1.5 m/s when it touches the ground for the first time. The points in this area are not included in the scope of this paper.

As can be seen from the white area boundary in Figure 10, the horizontal velocity boundaries that do not cause the fuselage to overturn are 1.728 m/s and −1.728 m/s when the slope angle is 0°. The horizontal velocity boundaries that do not cause the fuselage to overturn are about 1.694 m/s and −4.121 m/s when the slope angle is 20°. The positive horizontal velocity boundary is the smallest (about 1.492 m/s) when the slope angle is 10°. The negative horizontal velocity boundary is the smallest (about −10.037 m/s) when the slope angle is 8.5°.

When the horizontal velocity is positive and $V_Y \geq V_Z \cdot \tan \alpha$, the relative movement trend is along the slope upward when the landing gear touches the ground. The direction of friction $F_{\mu y}$ acting on the footpad is along the slope downward and the direction of normal force $F_n$ is perpendicular to the landing surface upward. The force analysis of the center of gravity can be obtained:

$$
\begin{cases}
M_X = f(V_Y, \alpha) \\
F_Z = f(V_Y, \alpha) \\
F_Y = f(V_Y, \alpha)
\end{cases}
\tag{6}
$$

Apparently $M_X > 0$, $F_Z < 0$, $F_Y < 0$ in current situation. The force $F_Z$ causes an acceleration along the negative direction to the rotorcraft along the $Z_b$-axis, which causes the rotorcraft to bounce. The force $F_Y$ causes an acceleration along the negative direction along the $Y_b$-axis. The torque $M_X$ causes an angular acceleration and makes the rotorcraft roll. After the landing gear is off the ground, the rotorcraft is only affected by gravity and maintains a certain roll angle velocity. Because of the roll angle, the No. 3 and No. 4 landing legs touch the ground before the No. 1 and No. 2 landing legs when the rotorcraft falls again due to gravity and touches the ground. At this time, the moment generated by

gravity relative to the footpad of the No. 3 and No. 4 legs causes the fuselage to produce a negative roll angular acceleration, thereby reducing the roll angular velocity and slowly increasing the roll angle to an extreme value. If the extreme value of the roll angle is not greater than $\tan^{-1}(L/(H - \Delta H))$, which means that the minimum stability distance is greater than 0, the rotorcraft will not overturn and stabilize under the influence of gravity.

When the slope angle $\alpha$ is fixed and the horizontal velocity $V_Y$ increases, the roll angular velocity leaving the ground increases, and the time interval from leaving the ground to falling again increases. Therefore, when the slope angle is fixed and the horizontal velocity increases, the roll angle and roll angular velocity when landing on the ground again increase, and the negative rolling moment generated by gravity cannot make the roll angle tend to be stable. The positive horizontal velocity stability boundary with the same slope angle is generated.

When the horizontal velocity $V_Y$ is fixed, and the slope angle $\alpha$ increases in the range of 0–20°, the roll angular velocity leaving the ground increases, and the time interval from leaving the ground to falling again increases slightly and then decreases rapidly. Therefore, when the horizontal velocity is fixed and the slope angle is small, the time interval is long, but the roll angle and roll angular velocity are small, which may allow the moment generated by gravity to stabilize the rotorcraft. When the horizontal velocity is fixed and the slope angle is large, the roll angular velocity is large and the time interval is short, which may produce a small roll angle and allow the moment generated by gravity to stabilize the rotorcraft. When the horizontal velocity is fixed and the slope angle is a certain value in the range of 0–20°, the roll angular velocity and the time interval may make the rotorcraft touch the ground with a large roll angle, and the moment generated by gravity cannot stabilize the rotorcraft. The slope angle stability boundary with the same positive horizontal velocity is generated.

When the horizontal velocity is positive and $V_Y < V_Z \cdot \tan \alpha$, the relative movement trend is along the slope downward when the landing gear touches the ground. The direction of friction $F_{\mu y}$ acting on the footpad is along the slope upward and generates a negative moment along the $X_b$-axis, which reduces the roll angular velocity. The range of horizontal velocity is relatively small and has a small impact on the movement of rotorcraft. Therefore, different combinations of slope angle and horizontal velocity allow the rotorcraft to land stably when $V_Y < V_Z \cdot \tan \alpha$. The landing stability boundary of slope angle and positive horizontal velocity is shown in Figure 10a.

When the horizontal velocity is negative, $M_X < 0$, $F_Z > 0$, $F_Y > 0$. The force $F_Z$ causes an acceleration along the negative direction to the rotorcraft along the $Z_b$-axis, which causes the rotorcraft to bounce. The force $F_Y$ causes a positive acceleration along the positive direction along the $Y_b$-axis. The torque $M_X$ causes a roll angular acceleration and makes the rotorcraft roll.

When the horizontal velocity $V_Y$ is fixed, and the slope angle $\alpha$ increases in the range of 0–20°, the roll angular velocity leaving the ground decreases, and the time interval from leaving the ground to falling again decreases. Therefore, when the horizontal velocity is fixed and the slope angle increases in the range of 0–20°, the roll angle and roll angular velocity when landing on the ground again decrease and allow the moment generated by gravity to stabilize the rotorcraft. The slope angle stability boundary with the same negative horizontal velocity is generated.

When the slope angle $\alpha$ is fixed and the absolute value of horizontal velocity $V_Y$ increases, the roll angle velocity leaving the ground decreases, and the time interval from leaving the ground to falling again increases slightly and then decreases. Therefore, when the slope angle is fixed, and the absolute value of horizontal velocity increases, the roll angle and roll angular velocity when landing on the ground again may be too large to allow the moment generated by gravity to stabilize the rotorcraft. The negative horizontal velocity stability boundary with the same slope angle is generated. The landing stability boundary of slope angle and negative horizontal velocity is shown in Figure 10b.

Thus, the landing stability boundary in different landing slope angles and horizontal velocities is obtained, as shown in Figure 10, which can be used as a theoretical basis for subsequent landing judgment and landing control research.

## 5. Conclusions

In this paper, a new four-legged adaptive landing gear is designed to meet the landing stability requirements of multirotor UAVs in complex terrain conditions, and a mathematical dynamic model of adaptive landing gear is established in MATLAB/Simulink. Conclusions are drawn as follows:

A four-legged adaptive landing gear with a bionic cushioning structure is designed, which increases the passive cushioning capacity compared with the existing adaptive landing gears.

The whole aircraft drop test simulations in different terrain models are carried out, which verify adaptive landing ability in complex terrain conditions. Compared with the simulation data in the commercial software Simcenter3D, the error of impact force acting on the rotorcraft is within 7%. The errors of steady-state values of cushion stroke and roll angle are also maintained at a small value, which verifies the accuracy of the mathematical model.

On the basis of a landing stability criterion, the landing stability simulations in different working conditions with different landing structures are carried out. The feasibility of the landing stability criterion is verified by simulation data, and it is proved that the adaptive landing gear has better landing stability than the conventional landing gear.

The terrain slope angle and initial horizontal speed were selected as the main parameters affecting the landing stability, and the landing stability with different combinations of the two parameters is simulated. The maximum slope angle at which the landing gear could stably land is 20°. The maximum positive horizontal velocity boundary is 1.728 m/s, and the corresponding slope angle is 10°. The minimum negative horizontal velocity boundary is −10.037 m/s, and the corresponding slope angle is 8.5°.

**Author Contributions:** Conceptualization, X.N. and X.W.; methodology, Q.Y. and X.W.; validation, X.N. and P.Z.; formal analysis, X.N. and Q.Y.; investigation, P.Z.; resources, Q.Y., X.W., and H.N.; writing—original draft preparation, X.N. and P.Z.; writing—review and editing, X.N., Q.Y., and X.W.; visualization, X.N.; supervision, X.W.; project administration, Q.Y., X.W., and H.N. All authors have read and agreed to the published version of the manuscript.

**Funding:** This study was supported by the National Natural Science Foundation of China (No. 51905264), the Fundamental Research Funds for the Central Universities (No. NT2021004), the Aeronautical Science Foundation of China (No. 202000410520002), the Fund of Prospective Layout of Scientific Research for NUAA (Nanjing University of Aeronautics and Astronautics), the China Postdoctoral Science Foundation Funded Project (No. 2021M691565), the National Defense Outstanding Youth Science Foundation (No. 2018-JCJQ-ZQ-053), and the Priority Academic Program Development of Jiangsu Higher Education Institutions.

**Institutional Review Board Statement:** Not applicable.

**Informed Consent Statement:** Not applicable.

**Data Availability Statement:** Not applicable.

**Conflicts of Interest:** The authors declare no conflict of interest.

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
