# Peer review of "Research on Landing Stability of Four-Legged Adaptive Landing Gear for Multirotor UAVs"

_aerospace, doi:10.3390/aerospace9120776_

Round 1

Reviewer 1 Report

The article proposed a new four-legged landing gear of multi-rotor UAV with passive cushioning structure. In the landing gear each structure imitates the human legs.

 The four-leg landing gear is designed for topographic of complex terrains, but is simulated only for simplified landing terrain with height difference or slope angle.

 A control strategy of landing legs’ adjustment is proposed under these conditions.

A simple geometrical stability criterion for landing is used to avoid rotorcraft overturner.    

In text line 169: There is a typographical error: different font type in phrase

In text line 174: Reference to the Figure 5 is loss.

In text line 265: There is a typographical error: different font type symbol ‘α

In text line 272: There is a typographical error: in phrase different font type in phrase ‘points on the X0Y’

In text line 277: There is a typographical error: symbol ‘。

In text line 365: There is a typographical error: different font type symbol ‘α

In text line 318: Reference to the Figure 9 is loss.

In text line 339: Reference to the Figure 9 is loss.

The paper presents appropriate references.

Tables and figures in this paper make it easier understand data presented.

Author Response

Dear reviewers

Thank you for your letter and comments concerning our manuscript entitled “Research on Landing Stability of Four-legged Adaptive Landing Gear for Multi-Rotor UAV”. Those comments are valuable and very helpful. We have read through comments carefully and have made corrections. Based on the instructions provided in your letter, we uploaded the file of the revised manuscript. Revisions in the text are shown using red highlight and the responses to your comments are presented following.

We would love to thank you for allowing us to resubmit a revised copy of the manuscript and we highly appreciate your time and consideration.

Response to Reviewer’s Comments

Point 1: In text line 169: There is a typographical error: different font type in phrase

In text line 174: Reference to the Figure 5 is loss.

In text line 265: There is a typographical error: different font type symbol ‘α’

In text line 272: There is a typographical error: in phrase different font type in phrase ‘points on the X0Y’

In text line 277: There is a typographical error: symbol ‘。’

In text line 365: There is a typographical error: different font type symbol ‘α’

In text line 318: Reference to the Figure 9 is loss.

In text line 339: Reference to the Figure 9 is loss.

Response 1: Thank you very much for pointing out these typographical errors. We have corrected these errors in revised copy of the manuscript. As we have revised the manuscript, the line numbers are now changed, so we list the corresponding line numbers as follow:

line 169 – line 178

line 174 – line 185

line 265 – line 276

line 272 – line 283

line 277 – line 288

line 318 – line 332

line 339 – line 353

line 365 – line 378

Reviewer 2 Report

The document entitled "Research on Landing Stability of Four-legged Adaptive Landing Gear for Multi-Rotor UAV" is evaluated. In general the document is well structured and organized and the methodological process is complete and adequate.

It is recommended to review the following:

1. Between lines 37 to 50 there are several value judgments about other works, sentences such as "but poor adaptability to complex terrains", "but structure was relatively complex" or "but its structure was complex and its versatility was poor" should be avoided. While the limitations of previous work cannot be denied, judgments should not be made about it for the sake of knowledge growth.

2. The cross-references in the paper on lines 174, 318, and 339 should be revised as the following sentence appears: "Error! Reference source not found."

Author Response

Dear reviewers

Thank you for your letter and comments concerning our manuscript entitled “Research on Landing Stability of Four-legged Adaptive Landing Gear for Multi-Rotor UAV”. Those comments are valuable and very helpful. We have read through comments carefully and have made corrections. Based on the instructions provided in your letter, we uploaded the file of the revised manuscript. Revisions in the text are shown using red highlight and the responses to your comments are presented following.

We would love to thank you for allowing us to resubmit a revised copy of the manuscript and we highly appreciate your time and consideration.

Response to Reviewer’s Comments

Point 1: Between lines 37 to 50 there are several value judgments about other works, sentences such as "but poor adaptability to complex terrains", "but structure was relatively complex" or "but its structure was complex and its versatility was poor" should be avoided. While the limitations of previous work cannot be denied, judgments should not be made about it for the sake of knowledge growth.

Response 1: Thanks for pointing this out, we're sorry for using some inappropriate statements when introducing other works and have corrected the statements. The relevant paragraph (line 37-50) has been amended and marked in red in the manuscript which is attached to the response.

Point 2: The cross-references in the paper on lines 174, 318, and 339 should be revised as the following sentence appears: "Error! Reference source not found."

Response 2: Thank you for pointing out the typographical errors in this paper and we have corrected these errors on lines 174, 318, and 339 (lines 185, 332, and 353 now).

Reviewer 3 Report

     The article ‘Research on Landing Stability of Four-legged Adaptive Landing Gear for Multi-Rotor UAV’.

     Thank you for the opportunity to review this paper. I read it with interest. The study focused on the analysis of a four-legged multi-rotor UAV landing gear with a passive shock-absorbing structure. The conditions of the analysis were driven by the technical conditions of the UAV and the adopted landing terrain conditions and structures, as well as adaptive adjustments. The proposed mathematical model of the landing gear dynamics was solved using MATLAB/Simulink and compared with results from Simcenter3D, where its applicability was confirmed. A landing stability limit was found for various parameter combinations.

     The paper represents a very good scientific level, and the research on topics presented in it can be useful in the future analysis of the subject. All analyses look very good, however, I suggest you consider making the following adjustments:

1. In Figure 2 or another supplementary figure, the force vectors described in equations 1-4 should be shown. this will allow the correctness of the mathematical model to be assessed.

2. Figure 5 should also show the force vectors in addition to the component designations and dimensions.

3. The dimensional characteristics of the UAV and the boundary conditions of the calculations were missing.

4. The adaptation process has not been detailed, no adjustment conditions and ranges are given.

5. Does the 'bending' of the end member in contact with the ground during adaptation cause the UAV to skid or not on an inclined ground?

6. Lines 174; 318 and 339 - Error! Reference source not found.

Author Response

Dear reviewers

Thank you for your letter and comments concerning our manuscript entitled “Research on Landing Stability of Four-legged Adaptive Landing Gear for Multi-Rotor UAV”. Those comments are valuable and very helpful. We have read through comments carefully and have made corrections. Based on the instructions provided in your letter, we uploaded the file of the revised manuscript. Revisions in the text are shown using red highlight and the responses to your comments are presented following.

We would love to thank you for allowing us to resubmit a revised copy of the manuscript and we highly appreciate your time and consideration.

Response to Reviewer’s Comments

Point 1: In Figure 2 or another supplementary figure, the force vectors described in equations 1-4 should be shown. this will allow the correctness of the mathematical model to be assessed.

Response 1: Thank you for pointing out the lack of description of the force vectors in equations 1-4 and it does obscure the mathematical model, so we plot the force vectors in a new figure and named it as Figure 3, in the hope of describing the modeling process more clearly.

Point 2: Figure 5 should also show the force vectors in addition to the component designations and dimensions.

Response 2: To show the component designations and dimensions clearly, we have marked them in the new Figure 3. And the new Figure 3 is consistent with the simplified diagram of the aircraft used in the original Figure 5 (Figure 6 now), so the force vector, component designations and dimensions required in the original Figure 5 (Figure 6 now) can be obtained according to the new Figure 3. At the same time, since the original Figure 5 (new Figure 6) has been annotated a lot, in order to keep the display clean, we have not modified the original Figure 5 (Figure 6 now) for the time being.

Point 3: The dimensional characteristics of the UAV and the boundary conditions of the calculations were missing.

Response 3: Table 1 (behind line 119) and Figure 3 have been added to detail the mass of each component and the component designations and dimensions. Between line 324-330, we have added a description of the boundary conditions for landing stability searching. Due to structural limitations, the maximum adaptable slope angle is ±20° (according to line 172-185), and the range of initial horizontal speed is -15 - 15 m/s (most rotorcrafts’ cruising speed does not exceed 20 m/s). Due to space constraints, the figure only shows the horizontal speed range of -10.5 - 2 m/s and the variation tendency of the range not shown is the same.

Point 4: The adaptation process has not been detailed, no adjustment conditions and ranges are given.

Response 4: On line 172-185 the adaptive adjustment process used in this paper has been added. The four legs only adjust the coordinates relative to the Z axis of the body axes to adapt to different terrain, and the maximum adaptable height difference is ±0.1 m and the maximum adaptable slope angle is ±20°.

Point 5: Does the 'bending' of the end member in contact with the ground during adaptation cause the UAV to skid or not on an inclined ground?

Response 5: In this paper, we do not consider the skidding and ‘bending’ in simulation. When making the prototype, all components are made of metal and reinforced with 3D prints. The motors are restrained by brake pads and the ball hinge between the foot-pad and the leg will install additional parts such as springs so that it does not produce undesired rotation. In addition, a piece of rubber sheet is stuck onto the bottom of the foot-pad to ensure the contact with the ground. Based on these considerations, we think the ‘bending’ of components or skidding may not be considered in the simulation.

Point 6: Lines 174; 318 and 339 - Error! Reference source not found.

Response 6: Thank you for pointing out the typographical errors in this paper and we have corrected these errors on lines 174, 318, and 339 (lines 185, 332, and 353 now).

Round 2

Reviewer 1 Report

I thank you for the corrections made.

Reviewer 2 Report

I thank the authors for taking the recommendations in consideration. This reviewer considers the document ready for publication.

Reviewer 3 Report

     The article ‘Research on Landing Stability of Four-legged Adaptive Landing Gear for Multi-Rotor UAV’.

      Thank you for giving me the opportunity to read the manuscript again after the corrections made. All my comments have been taken into account and are supported by a relevant comment from the Authors in their response. The revised version of the manuscript has gained scientific value and is more communicative for the potential audience.